# Differential Transcriptional Reprogramming by Wild Type and Lymphoma-Associated Mutant MYC Proteins as B-Cells Convert to a Lymphoma Phenotype

**DOI:** 10.3390/cancers13236093

**Published:** 2021-12-03

**Authors:** Amir Mahani, Gustav Arvidsson, Laia Sadeghi, Alf Grandien, Anthony P. H. Wright

**Affiliations:** 1Division for Biomolecular and Cellular Medicine, Department of Laboratory Medicine, Karolinska Institutet, 17177 Stockholm, Sweden; mahani_amir@hotmail.com (A.M.); gustav.arvidsson@medsci.uu.se (G.A.); laia.sadeghi@ki.se (L.S.); 2Center for Hematology and Regenerative Medicine, Department of Medicine, Karolinska Institutet, 17177 Stockholm, Sweden; grandien1234@gmail.com

**Keywords:** MYC, MYC mutants (T58A and T58I), lymphoma, oncogene deregulation, cell cycle, ribosome biogenesis, DNA replication

## Abstract

**Simple Summary:**

Oncogenic transformation in response to the activation of MYC requires secondary events that reduce the role of MYC as an inducer of apoptosis and cellular senescence. We made a primary B-cell model in which such secondary processes were reduced and MYC levels could be regulated at the transcriptional level. We studied the changes in transcriptional programs as MYC levels were progressively increased, causing a simultaneous transition to a lymphoma cell phenotype. Two lymphoma-associated MYC mutants (resulting in the substitution of threonine 58 with isoleucine, T58I, or alanine, T58A) were also analyzed and compared with each other and wild type MYC. All three MYC proteins induced cell cycle entry, cell growth and cell proliferation with associated changes in gene expression programs. The MYC-regulated genes overlap with MYC target genes in B-cells during normal activation. A minority of the MYC-regulated genes (enriched in specific aspects of the cell cycle) were regulated significantly differently by the mutant MYC proteins compared to wild type MYC and in most cases between the two mutant MYC proteins. These genes did not overlap significantly with the set of genes regulated during B-cell activation, and may be specific to oncogenesis. Many target genes were more sensitive to regulation by T58A and less sensitive to T58I, compared to wild type MYC, as was also seen for the measured phenotypic processes. This, taken together with the different sets of genes that are differentially regulated by each mutant in relation to wild type MYC, indicates that the mutations augment the oncogenic activity of MYC by different mechanisms. The different effects of the mutations may result from the different effects they are predicted to have on the conformational properties of the intrinsically disordered region of the MYC protein that surrounds the threonine 58 residue.

**Abstract:**

The MYC transcription factor regulates a vast number of genes and is implicated in many human malignancies. In some hematological malignancies, MYC is frequently subject to missense mutations that enhance its transformation activity. Here, we use a novel murine cell system to (i) characterize the transcriptional effects of progressively increasing MYC levels as normal primary B-cells transform to lymphoma cells and (ii) determine how this gene regulation program is modified by lymphoma-associated MYC mutations (T58A and T58I) that enhance its transformation activity. Unlike many previous studies, the cell system exploits primary B-cells that are transduced to allow regulated MYC expression under circumstances where apoptosis and senescence pathways are abrogated by the over-expression of the Bcl-xL and BMI1 proteins. In such cells, transition from a normal to a lymphoma phenotype is directly dependent on the MYC expression level, without a requirement for secondary events that are normally required during MYC-driven oncogenic transformation. A generalized linear model approach allowed an integrated analysis of RNA sequencing data to identify regulated genes in relation to both progressively increasing MYC level and wild type or mutant status. Using this design, a total of 7569 regulated genes were identified, of which the majority (*n* = 7263) were regulated in response to progressively increased levels of wild type MYC, while a smaller number of genes (*n* = 917) were differentially regulated, compared to wild type MYC, in T58A MYC- and/or T58I MYC-expressing cells. Unlike most genes that are similarly regulated by both wild type and mutant MYC genes, the set of 917 genes did not significantly overlap with known lipopolysaccharide regulated genes, which represent genes regulated by MYC in normal B cells. The genes that were differently regulated in cells expressing mutant MYC proteins were significantly enriched in DNA replication and G2 phase to mitosis transition genes. Thus, mutants affecting MYC proteins may augment quantitative oncogenic effects on the expression of normal MYC-target genes with qualitative oncogenic effects, by which sets of cell cycle genes are abnormally targeted by MYC as B cells transition into lymphoma cells. The T58A and T58I mutations augment MYC-driven transformation by distinct mechanisms.

## 1. Introduction

The transcription factor MYC plays a central role in the onset and progression of several B-cell lymphomas [1]. The defining characteristic of Burkitt’s lymphoma (BL) is a translocation event where *MYC* is fused to the immunoglobulin heavy chain locus leading to its aberrant expression (t8; 14) (q24; q32) [2]. Less frequently in BL, *MYC* is translocated to one of two loci encoding for immunoglobulin light chains, but the result is nevertheless deregulated expression and a BL phenotype. Aberrant MYC expression is, however, not sufficient to trigger malignant transformation and secondary events, such as concordant over-expression of Bcl-xL (encoded by BCL2L1) and BMI1 that, for example, inhibit the intrinsic apoptotic pathway and the ARF/TP53 axis, respectively, are required for MYC-driven B-cell transformation [3].

In addition to its translocation, *MYC* is also the most frequently mutated gene in BL [4]. The conserved MYC homology box 1 (MB1) within the transactivation domain contains a transiently structured proline-rich sequence where missense mutations frequently occur in BL and other hematological malignancies [5,6,7,8,9]. Among these, mutations most commonly lead to a threonine to isoleucine substitution at residue 58 (T58I), while the somewhat less frequently occurring substitution with alanine at the same residue (T58A) mutation causes higher transforming activity than both wild type (WT) and T58I in cellular model systems [10]. Additionally, T58A has been shown to promote B-cell lymphomagenesis in murine models [10,11,12]. MYC stability is affected by sequential phosphorylation at S62 and T58, where the former activates and stabilizes MYC, while subsequent phosphorylation at T58 primes MYC for proteasome-mediated degradation [13]. Proteins carrying the T58A mutation have been shown to be stabilized while MYC proteins carrying T58I are surprisingly similar to WT in terms of half-life, indicating that additional mechanisms affecting MYC protein levels are affected when the threonine residue at position T58 is converted to other amino acids that cannot be phosphorylated [5,10].

MYC/MAX heterodimers bind motifs in the regulatory regions of genes with different affinity and thereby control the expression level of genes. While the heterodimer has the highest affinity for canonical E-box motifs (CACGTG) or variants thereof, MYC/MAX has been shown to bind to the regulatory regions of a large number of regulated target genes, including genes with non-canonical E-box motifs or without recognizable E-box motifs in regulatory regions [14,15,16]. The differing affinity of MYC/MAX for different E-box motifs associated with different genes allows for the specific regulation of target genes in a MYC level-dependent fashion, where the promoters containing high-affinity motifs are preferentially bound at low MYC levels and genes with progressively lower affinity are progressively engaged as MYC levels increase [16]. In a transformed cell with deregulated MYC expression, MYC would be capable of regulating an abnormally large set of genes, ranging from those for which MYC has the highest promoter affinity to those with much lower affinity for MYC. Such genes would be expected to include normal MYC target genes, as has been recently suggested [17], but also genes not normally activated by MYC because physiological MYC levels are too low to activate such genes.

Previous studies of global gene expression changes upon MYC over-expression have focused on gene expression changes in relation to MYC occupancy in regulatory gene regions [18,19] or in relation to E-box occurrences in regulatory regions upon gradually increasing MYC levels [16]. An in vivo study of the Eµ-Myc mouse model of B-cell lymphoma using B-cells at three different stages of lymphomagenesis, each with different MYC levels, identified almost four thousand significantly changed genes between cells representing the different stages [18]. Efforts have also been made to identify genes that are directly regulated by MYC by combining gene expression profiling with chromatin immunoprecipitation in a B-cell line [20], or, more recently, identifying a list of 100 direct MYC targets and subsequently validating these targets across an array of human cancers by employing techniques for rapid induction of MYC degradation in combination with RNA-seq of nascent RNA [21]. Another study investigated global gene expression changes following the over-expression of MYC carrying mutations in MBI in relation to WT MYC in rat fibroblasts and found significant differences in gene expression for hundreds of genes between cells expressing WT MYC and the MYC mutants [22]. However, in that study, it was difficult to identify functions and pathways differentially regulated by the MYC mutants in relation to WT, possibly due to the cellular context. The study did, however, describe a differential regulation of Nop56 and a subsequent differential transformation capacity between cells overexpressing WT MYC and T58I, suggesting that subtle alterations to the sequence and protein conformation of MYC can have large effects on gene regulation and phenotypic manifestations.

In the present study, we use a transduced primary B-cell model of MYC-driven B-cell lymphoma to identify genes that are regulated as essentially normal murine B cells that transform into lymphoma-like cells in response to progressively increasing MYC levels and further to identify how the lymphoma-associated T58A and T58I mutations affect this transition in terms of altered gene expression programs. Anti-oncogenic apoptotic and senescence pathways were abrogated in the cells, causing them to directly transit to lymphoma cells in response to deregulated MYC without the need for other oncogenic events. Indeed, previous studies have shown that analogously modified cells cause lymphomas when injected into mice [23].

## 2. Results

### 2.1. Effects of Progressively Increased Wild Type and Mutant MYC Levels on Cell Growth, Proliferation and Cell Cycle Distribution

An in vitro cell model for investigating the role of MYC in lymphoma development was developed using LPS-activated primary murine B-cells, transduced with retroviral vectors containing doxycycline-regulated coding sequences for wild type (WT) MYC or one of two lymphoma-associated MYC mutants, in which threonine-58 is substituted with alanine (T58A) or isoleucine (T58I), together with vectors for constitutive expression of the anti-apoptotic proteins Bcl-xL, encoded by *BCL2L1*, and BMI1, the latter also inhibiting cellular senescence, as described previously [3]. Concordant expression of the three proteins is necessary for the oncogenic transformation of the murine B-cells, as they otherwise undergo apoptosis or fail to proliferate. The resulting transformed cells expressed B-cell markers (CD19 and CD45R), but were negative for CD90 (expressed in T-cells but not B-cells), indicating that the transduced cells were predominantly B-cells as expected (Appendix A).

To understand changes in cellular characteristics and how they differ between WT and mutants as increasing MYC levels drive the transduced B-cells towards a lymphoma phenotype, we overexpressed MYC, T58A and T58I at different levels by progressively increasing the doxycycline concentrations added to the transduced cells for 48, 96 and 144 h (Figure 1A). The doxycycline-mediated increase in WT and mutant MYC proteins is shown in Figure 1B (see also Appendix A for a representative Western blot image, and Appendix A for original Western Blot data). Consistent with previous studies, the T58A mutant reaches somewhat higher levels than the WT and T58I mutant proteins, which are expressed at similar levels [5,10]. As shown in Figure 1C, cells can be seen to enter the cell cycle by 48 h (fewer G_1_/G_0_ cells and more S-phase cells), but there is little or no change in cell growth or proliferation (cell size or number). At later time points (96 and 144 h), the cells progressively grow (larger size), proliferate (increased number) and enter the cell cycle in response to both MYC level and time. T58A cells outperform WT and T58I cells in these respects by showing greater sensitivity to the doxycycline level, presumably due, at least in part, to the higher expression level of the T58A mutant protein. Interestingly, the T58I mutant shows the lowest response sensitivity with regard to growth, proliferation and cell cycle entry both in relation to MYC level and time.

### 2.2. Global Gene Expression Changes in Response to Progressively Increased Levels of Wild Type and Mutant MYC Proteins

To identify MYC level-associated changes in transcript levels and the differences between WT and mutants at a global scale, mRNA was extracted following 48 h of MYC over-expression for seven different levels of MYC expression. This time point precedes measurable changes in cell size or number but coincides with phenotypic signs of cell cycle entry. We would thus expect to observe regulation of genes involved in MYC-induced lymphoma development at 48 h, even though some genes are likely to be indirect targets of MYC. While differences in global transcript levels were observed between the B-cells based on *MYC* mutation status, as visualized by the principal component analysis in Figure 2A,B, the largest effect is related to MYC level, where there is a consequent shift to the right for principal component 1 as MYC levels increase, while principal components 2 and 3 reveal differences between genotypes. The expression dataset was subsequently subject to the glmQLF generalized linear model workflow within the EdgeR package to extract gene models with significant transcript level changes in WT in response to increasing MYC levels, as well as for genes that were differentially changed between WT and T58A or T58I. In response to increasing MYC levels, we found 7263 (WT), 347 (WT vs. T58A) and 683 (WT vs. T58I) significant gene models for the respective comparative groups. Selected models were required to have a two or more-fold changed transcript level between at least two conditions (MYC level or mutant status) and an FDR q-value < 0.01 (listed in Appendix A).

### 2.3. Clusters of Gene Models Describe Genes with Similar Patterns of Differential Gene Expression

Agglomerative hierarchical clustering was then used to identify clusters of similar gene models within the combined group of selected gene models, based on mean log_2_ transformed RPKM values for the different MYC levels, for WT and mutant MYC proteins independently (Figure 3). The resulting twelve clusters contained between 55 and 1455 genes and could be divided into two cluster classes depending on whether the overall trend in transcript levels was increasing (clusters c3, c7, c9, c10, c11 and c12) or decreasing (clusters c1, c2, c4, c5, c6, and c8) in response to increasing MYC levels (Appendix A). As is evident in Figure 3, some clusters contained genes characterized by high transcript levels (e.g., clusters c5, c7 and c9), while others were characterized by genes with low transcript levels (e.g., clusters c4 and c11). Five clusters were significantly enriched for non-coding genes (c2, c6, c8, c11 and c12), while for six clusters, a significant under-representation was observed (c3, c4, c5, c7, c9 and c10). Notably, cluster c2, which contained the lowest number of genes and almost exclusively contained non-coding genes, also had significantly shorter genes when compared to non-regulated genes (median feature length: 366 base pairs (bp) compared to 3810 bp). These and other characteristics of the 12 clusters are summarized in Table 1.

Figure 3 also shows differences between the effects of WT and mutant MYC proteins, most notably with regard to T58A, which exhibits large changes for many genes, even at the lowest doxycycline concentration, where MYC levels are scarcely higher than in WT or T58I. We conclude that the significant gene models describing changes in the dataset identify gene expression changes in response to increasing levels of MYC as well as differences in changes between wild type and mutant MYC proteins and between the two mutant MYC proteins.

### 2.4. Close Proximity and Higher Number of Canonical E-Boxes near Transcription Start Sites Is Associated with Gene Clusters Characterized by Up-Regulated Genes

MYC/MAX heterodimers bind with high affinity to canonical E-box motifs (CACGTG) that are generally located closely upstream or downstream of transcription start sites (TSS), leading to the activation of transcription [16]. We identified the positions for all canonical E-boxes in the mouse reference genome and derived the distance of each to the nearest TSS. For clusters c3, c7, and c10, there was a significant enrichment of genes with both at least one canonical E-box within 1000 bp of the TSS and a significantly shorter median distance between the TSS and the closest canonical E-box compared to non-regulated genes (Table 1).

Each of these clusters contain a majority of genes that are up-regulated in response to increasing MYC levels. Conversely, cluster c2 and c6, characterized by genes with reducing transcript levels in response to MYC level, exhibited the opposite pattern compared to non-regulated genes (i.e., significantly fewer E-boxes ± 1000 bp from the TSS and significantly longer median distance between TSS and closest E-box). The enrichment of canonical E-boxes in clusters characterized by genes that are up-regulated in response to increasing MYC levels suggests that in these cells, MYC primarily acts as a positive regulator of target genes with high-affinity binding sites.

### 2.5. Clusters of Similar Gene Models Tend to Be Enriched in Genes with Distinct Functional Roles

We used gene ontology (GO) analysis to cross-validate the gene expression data in relation to the existence of logical patterns in independently gathered functional data [24]. The twelve clusters were subject to enrichment analysis for gene ontology (GO) term gene sets (C5, biological processes) and KEGG pathways (Figure 4, Appendix A as well as Appendix A). Enrichment of GO terms was found for all clustered gene sets (FDR adjusted *p*-value ≤ 0.1). Interestingly, there was little overlap between the clusters with regard to the significantly enriched GO terms. In cluster c1, enrichment was primarily found for a select number of down-regulated TOLL-like receptors for lipopeptides and genes with anti-apoptotic functions. Cluster c2 mostly contained non-coding RNA features which resulted in scarce enrichment of GO terms, presumably due to the poorer annotation of non-coding genes and the relatively small size of this cluster. In cluster c3, there was an enrichment of genes implicated in positive regulation of DNA repair and DNA replication as well as gene sets for regulation of the G_2_/M transition of the cell cycle. Cluster c4 genes show enrichment in several GO terms (also enriched in cluster c6) that cannot intuitively be coupled to B cells, but they also show enrichment in the “positive regulation of small molecule metabolic process” term. There are a large number of child terms (component GO terms) in this GO term, and it is not apparent which of them might be coupled to MYC in this context. Genes associated with the “cytokine–cytokine receptor interaction” KEGG term are also enriched in cluster c4. In cluster c5, enrichment was found for B-cell identity genes with functions in B-cell activation and differentiation, which are down-regulated. The enriched GO terms are not intuitively informative for cluster c6, although notably, this cluster is enriched in non-coding RNA transcripts, which are generally poorly annotated. The latter is also true for cluster c8 but here there is also an over-representation of down-regulated genes involved in chemotaxis. Cluster c7, predominantly containing up-regulated genes with high expression values, has an over-representation of genes involved in ribosome biogenesis as well as mitotic nuclear division (also enriched in cluster c10). Cluster c9, principally containing up-regulated genes, has an over-representation of genes associated with purine metabolism, while cluster c10 is enriched in genes implicated in non-coding RNA metabolic processes, RNA binding and mRNA transport as well as mitotic nuclear division. Cluster c11, mostly containing up-regulated genes, has a weak enrichment of genes involved in cAMP-mediated signaling. Cluster c12 does not contain genes enriched in processes intuitively related to B cells, but does show weak association with some other GO terms. Notably, clusters c11 and c12 contain about 25% non-coding RNAs, including lincRNAs, potential protein-coding transcripts that await validation (TEC) and other transcripts.

### 2.6. Comparison of Regulated Genes to Independently Identified MYC-Regulated Genes in an In Vivo Lymphoma Model and to Direct MYC-Target Genes

To further validate and characterize the cell model and the obtained gene expression results, we determined the extent to which the MYC level-associated genes identified in this cell system are related to genes regulated during lymphoma development in vivo by comparing our results with existing expression data for the Eµ-Myc mouse model of lymphoma [18]. In that study, Sabò et al. performed RNA-seq analysis (among other analyses) on isolated B cells at three stages of lymphomagenesis, each with different and progressively increasing MYC levels. We observed a large significant overlap (3991 genes, 55%) between the genes with significantly changed transcript levels in response to WT MYC in this study and the genes with significantly altered transcript levels in the in vivo study by Sabò et al. (Figure 5A), which corresponds to 68% of the regulated genes identified in the Eµ-Myc model data. In general, the genes regulated during lymphoma development in vivo were well represented throughout the 12 clusters that describe the data from the present study (Figure 5B). Notably, there was a significant over-representation of intersecting genes in clusters c3, c7, c9 and c10, which were functionally classified as containing genes involved in the cell cycle, ribosome biogenesis, purine metabolism and pyrimidine metabolism and RNA-related processes (Figure 4).

Using SLAM-seq to identify direct MYC targets (genes for which the level of nascent transcripts was changed after rapid reduction in MYC levels), Muhar et al. identified a 100 gene signature that correlated well with genes regulated in a wide range of human cancers characterized by highly de-regulated MYC and where high MYC expression is a negative prognostic factor [21]. Eighty-eight of the human signature genes had reliably detectable orthologues in mouse cells, and of these, 71 significantly overlapped with the set of significantly regulated genes identified in the present study (Figure 5C). There was also an enrichment of direct MYC target genes among the most significantly changed genes in the present study (Figure 5D). A similar enrichment was found for the overlap of differentially regulated genes from the present study and a list of 668 direct MYC target genes that was identified using a murine B cell line [20] (Appendix A).

The 71 genes that were common to the direct MYC target genes identified by Muhar et al. had canonical E-box motifs significantly closer to the TSS than non-regulated genes (median distance: 169 bp as compared to 965 bp for the non-regulated genes, resampling *p*-value = 0.0046). A significant enrichment for the 71 direct target genes was observed in the up-regulated clusters c7, c9 and c10 (Appendix A).

We conclude that the MYC level-associated genes identified in this study, and particularly the 1886 genes making up clusters c7, c9 and c10, cross-validate well in relation to genes that change during lymphoma development in the Eµ-Myc in vivo lymphoma model as well as human signature genes that are characteristic for a wide range of human cancers with highly de-regulated MYC expression and high MYC levels as a negative prognostic factor.

### 2.7. Genes That Respond Differently in Response to MYC Level in Cells Expressing Lymphoma-Associated MYC Mutants Compared to Wild Type MYC

A subset of MYC level-associated genes responded significantly differently to increasing levels of MYC, depending on the mutant status of the expressed MYC protein (WT, T58A or T58I, Figure 6A and Appendix A). For most genes, the differences in the MYC level-associated regulation pattern between cells expressing different MYC proteins do not reach statistical significance (*n* = 6652, Figure 6A subset A). However, 306 genes show significant changes in relation to the MYC level in one or both mutants, but not in the WT MYC cells (Figure 6A, subsets B, C and F). A further 611 genes are regulated in cells expressing all three proteins but differentially in one or both mutants compared to WT (Figure 6A, subsets D, E and G). It is notable that there are many more genes differently regulated in only one of the mutants individually compared to WT (Figure 6A, subsets B, C, D and E, *n* = 803) than there are genes that are differently regulated in both mutants, relative to WT (Figure 6A, subsets F and G, *n* = 113).

An enrichment analysis using the KEGG pathway database for each of the mutant-specific subsets in Figure 6A (sets B–G) yielded few significantly enriched functional categories of interest, plausibly due to the small size of the gene subsets (Appendix A). Instead, sets A–G were tested for gene intersections with clusters c1–c12 and significant over- and under-representations of genes were found for several intersections in relation to the intersection sizes expected for a random assortment of genes between clusters (Figure 6B). For example, for genes in Set G, where the MYC-level response of genes differed in both mutants relative to WT, there is an under-representation of growth- and proliferation-related genes represented by cluster c10, a cluster which is otherwise a strong over-representation of Set A genes, which are similarly regulated in cells expressing wild type and mutant MYC proteins. Conversely, Set G genes are over-represented in cluster c5 and c8 genes, which are characterized by down-regulated genes involved in, e.g., B-cell differentiation and chemotaxis, suggesting that the mutant MYC proteins show a significant propensity to affect the expression of genes in these clusters differently from the wild type.

The gene sets showing most over/under-representation in different clusters are those associated with differences compared to WT for each mutant individually (Figure 6B, Sets B, C, D and E). Interestingly, both mutants individually differentially regulate genes that are over-represented in cluster c3 (Sets D and E), a cluster which is enriched in genes involved in DNA replication and the entry into mitosis. Interestingly, A-subset genes, which do not exhibit significant differential regulation by the different MYC proteins, are significantly under-represented in cluster c3 genes. As for Set G genes, Set D and E genes are instead strongly underrepresented in cluster c10 genes, where Set A genes are strongly over-represented. Taken together, this suggests that the regulation of a group of genes important for mitotic entry and DNA replication tends to be affected by the substitution of threonine 58 with alanine or isoleucine, while cluster 10 genes, important for a range of growth and proliferation processes, tend not to be affected by the mutations.

Since the gene regulation program underlying lymphomagenesis involves the regulation of normal MYC target genes that are also required for MYC’s role in the growth and proliferation of normal lymphocytes [17], we reasoned that such genes might be overrepresented in Set A genes, while genes differentially regulated by mutant MYC proteins might represent a group of genes characterized by abnormal MYC target genes more specifically associated with lymphomagenesis. Consistently, there is a highly significant overlap between Set A genes and a group of MYC target genes regulated during normal lymphocyte activation identified previously [17] (*n* = 4140) (Figure 6C). Interestingly, the overlap between these genes and genes differentially regulated by one or both MYC mutants compared to wild type MYC was too small to reach statistical significance. This result supports the idea that substitution mutants affecting the MYC protein may augment its oncogenic role during lymphomagenesis by extending the repertoire of available MYC target genes.

### 2.8. Differences in MYC-Level Associated Gene Regulation between WT, T58A and T58I

The progressive occupancy of MYC binding sites with progressively reduced affinity in response to increasing levels of MYC has been shown to be important for understanding the normal function of MYC as well as its function in cancer progression [16]. As shown by Figure 3, it is evident that the T58A mutation increases the MYC response sensitivity of many genes. A gene set enrichment analysis (GSEA) was thus performed using a pre-ranked list based on the sensitivity of gene expression to the level of MYC in T58A compared to T58I (details can be found in Materials and Methods), together with gene sets from the KEGG pathway database, GO terms for biological processes and a list of curated hallmarks [25]. Twenty-nine significantly enriched gene sets were found (adjusted *p*-value ≤ 0.05 and absolute normalized enrichment score ≥ 1.7, Appendix A). For the majority of the selected gene sets, the sensitivity to the level of MYC was higher for T58A than for T58I, and these sets represented different aspects of the cell cycle and genome integrity processes. Only one set (biological process—peptide cross linking) was found where sensitivity was higher in the T58I mutant. Results for a selection of pathways with strong enrichment are presented in Figure 7.

The sensitivity of the T58I mutant protein is lower than WT and T58A with respect to the induction of cell cycle entry, cell growth and cell proliferation (Figure 1), and a similar pattern is seen for MYC-dependent regulation of many functionally relevant genes (Figure 3, Figure 4 and Figure 7B). Thus, the effect of T58I on these functions cannot easily account for its oncogenic role. As an alternative way to find genes with specific responses to MYC in T58I, we identified T58I-associated genes (subsets C, E, F, G) that differed from WT or T58A by ≥1.2-fold at an intermediate doxycycline concentration (300 ng/mL). Figure 8A shows examples of genes that reflect the predominant pattern in which T58I exhibits lower sensitivity compared to WT and T58A. Many of the selected genes represent functions required for cell cycle progression where the lower activity of T58I could contribute to its lower functionality in driving the cell cycle (see Figure 1). However, there are genes showing the opposite pattern in which T58I exhibits a higher level of activity than one or both the other MYC proteins (Figure 8B). The selected group contains signaling proteins and it links to the cell cycle, cell structure and the cell membrane/secretion system, as well as functions that have previously been studied in neural cells. Figure 8C contains genes that show an opposite pattern of MYC-dependent regulation in T58I compared to WT and T58A. Many of the genes are involved in aspects of metabolism, cell cycle or transcriptional regulation. For example, the MNT protein is a competitive inhibitor of MYC due to the formation of heterodimers at MYC binding sites with the Max protein. In T58I, MNT is down-regulated, perhaps serving to augment the level of MYC activity, while in WT and to a greater extent in T58A, it is up-regulated, perhaps in order to abrogate MYC activity. Taken together, the results show that there are a number of genes whose differential regulation by T58I compared to WT or T58I could contribute to T58I-specific oncogenic mechanisms, but it is not possible to show enrichment in specific functional categories of genes.

Different genes require the recruitment of different repertoires of co-activator and co-repressor proteins for their regulation, and the MBI region in the MYC N-terminus is known to be important for interactions with many MYC partner proteins, as well as for regulating MYC activity and stability [26]. Since flexible MYC conformation is important for coupled binding and folding mediated protein interactions [27,28], it is conceivable that the T58I mutation could affect the conformational properties of MBI, and thereby its interaction with all or a subset of MYC partner proteins. Indeed, we showed previously that T58I but not T58A is predicted to affect the level of disordered conformation in MBI [29]. Figure 8D shows that the backbone flexibility (Dynamine) and protein interaction propensity (Anchor) of MBI are predicted to be lower for T58I compared to WT and T58A. There is no effect of the mutations on the predicted alpha-helical propensity of a previously identified transiently alpha-helical region (residues 47–53) [9] immediate N-terminal to residue 58 in MBI (Helix Agadir).

## 3. Discussion

Here, we present and characterize the roles of MYC in B-cell lymphoma development in B cells that are primed for MYC-induced lymphoma onset by the over-expression of anti-apoptotic factors (BCL2L1 and BMI1) and that therefore directly and progressively adopt a lymphoma-like phenotype as MYC levels are increased. Analogous cells to those described here cause rapid lymphoma development in vivo [3]. The cell system benchmarks well against previous knowledge from many in vivo and in vitro studies that have identified genes targeted by over-expressed MYC. To our knowledge, this is the first study to identify MYC target genes in relevant B cells that directly convert to lymphoma cells in response to increasing wild type or mutant (T58A and T58I) MYC levels, due to the prior disruption of apoptosis- and cellular senescence-inducing pathways.

### 3.1. Genes Regulated in Response to Progressive Increase in Wild Type MYC

As would be expected during lymphoma onset, the progressive increase in MYC levels in this study was associated with increased cell growth, entry of the transduced cells into the cell cycle and cell proliferation with increased expression of relevant underlying genes. Concordantly, genes related to differentiated B-cell characteristics were down-regulated in response to increasing MYC levels. Notably, gene clusters, defined by MYC level-dependent changes in transcript levels and MYC genotype, showed few overlaps in terms of the functional categories for which enrichment of genes was found. This suggests that values for target gene expression level, the direction and sensitivity of MYC-mediated regulation and MYC genotype is sufficient, at least in this case, to define gene sets with different cellular functions. It is evident that target gene expression level weighs heavily in the clustering process (Figure 3), and this is therefore an important parameter defining gene function in this context. MYC level, associated with the sensitivity and direction of target gene response, also seems to be associated with different classes of genes as has also been observed in analogous studies in an osteosarcoma cell line [16].

MYC dysregulation is implicated in the onset and progression of several hematological malignancies, and its involvement ranges from being disease-defining, as in Burkitt’s lymphoma [30], to cases where increased MYC expression, by means of deregulation or amplification events, is correlated with an inferior clinical outcome [1]. Recent global gene expression studies have largely focused on identifying genes directly regulated by wild type MYC in relation to experimentally detected MYC binding sites or in relation to E-box sequences in P493-6 cells [20,31], U2OS cells [16,19], B- and T-splenocytes [32] or in the Eµ-Myc mouse model [18], collectively showing that MYC has the capacity to increase or decrease the expression of large and diverse, yet discrete sets of genes in different cellular backgrounds. The number of regulated genes upon increased MYC levels in this study (*n* = 7569) was comparable to previous results using the Eµ-Myc transgenic mouse model, and the large overlap with regulated genes in that study indicates that this in vitro system faithfully recapitulates major parts of the MYC-driven lymphomagenesis process observed in vivo [18].

Genes regulated in response to MYC over-expression across different model systems have identified common themes related to basic cellular functions such as ribosome biogenesis, RNA processing and biomass accumulation, suggesting a conserved role for MYC in these processes [33]. Indeed, the same processes are regulated by MYC in normal B-cells [17], as well as by the MYC orthologue in *Drosophila melanogaster* [34]. MYC has consistently been shown to be a direct regulator of gene transcription by RNA polymerases I and III, in addition to its role in protein-coding gene transcription, and has thus been suggested to be an overall coordinator of growth-related transcription [35,36,37]. Mice with either insufficient MYC or reduced ribosome biogenesis have a slower incidence of MYC driven lymphomas [38,39], and an inhibitor of RNA polymerase I transcription is in clinical trials in patients with hematological malignancies [40]. Consistent with previous studies, clusters c7 and c10 contain genes involved in ribosome biogenesis and rRNA/tRNA processing with predominantly increasing transcript levels in response to increasing MYC levels [18,22]. Interestingly, promoter regions for genes with functions in ribosome biogenesis also have the highest affinity for MYC binding [16], and in the present study, many such genes are highly induced at relatively low MYC levels.

Another study, using RNA-seq measurements of nascent RNA upon abrupt MYC depletion, found that 36% of all factors involved in ribosome biogenesis were directly regulated by MYC, as were regulators of adenosine 5′-monophosphate (AMP) metabolism and de novo purine synthesis [21]. These observations are consistent with the present study, where an enrichment of genes involved in purine metabolic processes and cAMP-mediated signaling are found in the up-regulated clusters c9 and c11, respectively [21]. Regulation of purine metabolism by MYC has been described before in a B-cell context [41]. The gene signature with direct MYC targets has been validated across 5583 patient samples representing a diverse set of human malignancies [21], and the observed strong overlap with the present study serves as another strong validation for this model system and the results obtained in relation to lymphomagenesis.

The observation that canonical E-boxes in the predominantly up-regulated clusters c3, c7, c9, c10 and c11 were more frequently close to the TSS of genes or were closer to the transcription start sites than for non-regulated genes was indicative of a likely direct regulation of many of these genes by MYC. This conclusion is further supported by the significant enrichment of the experimentally determined direct target genes [21] in clusters c7, c9 and c10. Cluster c5 is also enriched in genes with canonical E-boxes close to the TSS, but almost all genes in this cluster are down-regulated. Low-affinity E-boxes have previously been associated with genes that are down-regulated in response to activated MYC [16], but the potential role of MYC as a direct repressor of gene regulation remains to be conclusively established. The *p*-value for cluster c5 is close to the significance threshold, and thus our data do not contribute strongly to the resolution of this issue.

Taken together, the results confirm previous knowledge about gene programs regulated by wild type MYC in normal and cancer cells, and show that this knowledge also applies in the context of normal B cells primed for MYC-driven transformation to a lymphoma phenotype by prior ablation of pathways leading to the induction of apoptosis and cellular senescence.

### 3.2. Role of Lymphoma Associated MYC Mutations

Perhaps the most important result of this study is that substitution of T58 by alanine or isoleucine causes opposite phenotypic effects in this lymphoma model. The T58A mutant drives cell growth, cell cycle entry and cell proliferation more strongly than wild type MYC, while the opposite is true for the T58I mutation. At least part of the increased sensitivity of the T58A mutant is likely associated with its higher expression level, which probably results from increased protein stability resulting from the absence of T58 phosphorylation, as has previously been shown in other contexts. However, it is apparent that this mechanism is insufficient to account for the role of T58 mutations in lymphoma development. This is because (i) the T58I mutant is not expressed at a detectably higher level than the wild type, even though it cannot be phosphorylated at residue 58, and (ii) the resulting phenotype for T58I is the opposite of the T58A phenotype, indicating that different oncogenic mechanisms associated with the different substituting amino acids must be involved. The differences between T58A and T58I at the level of phenotype are further established at the level of gene regulation, both in terms of the largely distinct sets of genes that are differentially regulated by the mutants and at the level of the distinct effects of the mutations in relation to the wild type on the expression pattern of numerous genes. Interestingly, gene cluster c3 that is characterized by cell-cycle related functions, is over-represented in the set of genes that are differently regulated in T58A- or T58I-expressing cells compared to cells expressing WT MYC. Thus, both mutations appear to impact on cell cycle functions but via different sets of genes and mechanisms. In this respect, it is also important to note that the gene models for most regulated genes do not reveal statistically significant differences between the wild type and mutant MYC proteins.

While this type of systematic study of lymphoma-associated mutations has not been performed previously, some previous studies have been performed in rat fibroblast cell lines [10,21]. Consistently, these studies showed higher expression of T58A mutant proteins compared to T58I, which was similar in expression level to the wild type. This corresponded to a much larger effect of the alanine substitution on MYC protein stability compared to the isoleucine substitution that caused only a low level of stabilization. Similarly to the present study, the fibroblast-based studies showed that while many genes were regulated similarly in wild type and mutant MYC-expressing cells, there was also a large group of genes that were differentially regulated in mutant MYC expressing cells (T58I and E39D). Consistent with the present study, T58A was associated with a higher transformation frequency compared to wild type MYC, while T58I led to lower transformation frequency than the wild type. The difference may be associated with the reduced phosphorylation of S62 that is observed in this T58I but not T58A, since S62 phosphorylation is required for activation of MYC [10]. Thus, at least some aspects of the present results appear to reflect general characteristics of the mutant MYC proteins that also apply outside the lymphoma context.

### 3.3. Difference between T58A MYC and T58I MYC

It is important to note that in lymphoma, mutations of T58 are observed in the context of MYC proteins that are over-expressed due to genetic rearrangements. In many cases, such rearrangements lead to lymphoma in the absence of mutations in MYC. Amino acid substitution mutations such as T58A and T58I should therefore be regarded as oncogenic events that augment the effects of oncogenic genetic rearrangements. It has been noted previously in rat fibroblast cell lines [10] that most lymphoma-associated amino acid substitution mutations in MYC reduce its transformation activity in in vitro assays, and these include T58I, which is one of the most frequently occurring lymphoma-associated MYC mutations. Conversely, T58A is rarely observed in the clinic, but causes higher than wild type transformation activity. However, MYC is an enigmatic protein that induces processes that are positive for oncogenesis, such as cell growth, cell cycle entry and cell proliferation as well as negative processes, such as apoptosis and the induction of cellular senescence. The same study showed that T58I induced wild type levels of apoptosis, while T58A MYC induced lower levels of apoptosis. Consistently, the T58A mutation has previously been reported to uncouple the role of MYC as an inducer of cell growth and proliferation from its induction of apoptosis during lymphoma onset [12]. We could not meaningfully measure the effects on apoptosis or senescence in this study due to abrogation of these pathways in the cell model, but taken together with previous work, it seems that the uncoupling mechanism provides the most plausible explanation for the augmenting effect of the T58A mutation.

Since MYC induces processes that are both positive and negative for tumorigenesis, a second strategy for optimal tumorigenesis may be to establish a level of MYC activity that optimizes the balance between positive and negative processes. Indeed, there is evidence that positive process genes are activated by lower levels of MYC than negative process genes [42]. In a scenario where MYC is over-expressed at higher than optimal levels, there would be an opportunity for mutations that reduce MYC activity to be selected, thus optimizing its tumorigenicity. This would thus be one way to account for the common occurrence of reduced-activity mutations such as T58I in lymphomas, particularly those such as Burkitt’s lymphoma, where MYC activation is known to be the primary driver.

A further possible explanation for the role of T58I in tumorigenesis, which is not mutually exclusive to the previous one, is that T58I enhances the MYC-dependent activation of important tumorigenic processes, other than cell growth and proliferation, by activating a subset of genes more efficiently than WT MYC. For example, migration towards and adherence to stromal cells in microenvironments has been shown to enhance the survival properties of some lymphoma cells (see [43] and references therein). While T58I MYC is less active than WT and T58A MYC for the regulation of most genes, the group of genes that are differentially regulated in T58I compared to WT does contain a set of genes that are more efficiently regulated by T58I or where the direction of regulation differs in T58I compared to WT or T58A MYC. We were not able to show statistical support for the over-representation of genes with particular functions in this gene set, but the set contains genes involved in signaling and gene regulation processes as well as metabolism that potentially could be associated with appropriate functionality changes in lymphoma cells.

It was suggested previously that the T58A and T58I mutations may cause different changes to local protein conformation, leading to quantitative or qualitative changes in the interaction of MYC with partner proteins [10]. There is consistent evidence that the MYC N-terminus is a conformationally disordered region and that conformational changes accompany interactions with partner proteins as part of a coupled binding and folding interaction mechanism [9,27,28,44]. Recently, the stability of protein regions with transient alpha-helical conformation has been shown to be an important modulator of activity for this kind of protein domain, see [45] and references therein, but neither T58A nor T58I is predicted to affect the transient alpha-helical region immediate N-terminal of T58. However, the Dynamine predictor of protein backbone dynamics predicts an increased backbone rigidity in the T58I mutant that is not seen for T58A, and the Anchor predictor predicts a greater reduction in protein interaction propensity for T58I than for T58A. It is thus possible that the transient extended proline-rich structure reported previously between residues 55 and 63 [9], which is characterized by restrained backbone dynamics, is stabilized in T58I but not T58A, leading to a more rigid conformation with lower propensity for coupled binding and folding. This provides a hypothetical model for differential effects of isoleucine and alanine substitutions of T58 on qualitative or quantitative aspects of interactions between MYC and partner proteins.

## 4. Materials and Methods

### 4.1. Preparation of Mouse Primary B-Cells

Ethical permission for the use of mouse primary B-cells was obtained from the Stockholm Norra Djurförsöksetiska Nämnd (Dnr. N375/12). Following local and national guidelines, splenic cells from C57BL/6 mice were prepared and stimulated with 25 µg/mL of LPS (Sigma-Aldrich, St. Louise, MO, USA) in RPMI medium with supplements, as described previously [3]. Generally, the percentage of B cells in the freshly isolated splenocyte populations is around 45% and the transduction frequency of mitogen activated B-cells is close to 50%. Obtained RNA sequencing data for the EGFP transcript, commonly expressed by each of the transduced cell populations, shows that EGFP is expressed slightly lower in the cell populations expressing mutant MYC proteins compared to wild type (T58A 0.57-fold; T58I 0.45-fold). This may indicate a slightly lower transduction efficiency for constructs expressing mutant MYC proteins in relation to the wild type.

### 4.2. Retrovirus Vector Construction

Vectors containing human wild type *MYC*, *T58AMYC* and *T58IMYC* were generously provided by M.D Cole (Department of Genetics, Geisel School of Medicine, Dartmouth, NH, USA). Human *WT MYC*, *T58A* and *T58I* sequences were amplified by PCR using SK primers containing *MLuI* digestion sites and cloned into the *MLuI*-cleaved pSIR-TRE-IRES-EGFP-PGK1-rtTA2 retrovirus expression vector (generously provided by Kari Högstrand), generating pSIR-TRE-MYC-IRES-EGFP-PGK1-rtTA2 vectors. The constructs were confirmed by DNA sequencing. The BCL2L1 and BMI1 retroviral expression vectors have been described previously [3]. In the present study, the vector containing *BCL2L1* (encoding for Bcl-xL) encoded EGFP in place of DsRed-Monomer.

### 4.3. Retroviral Transduction

Phoenix-Eco packaging cells (kindly provided by GP. Nolan, Stanford University, maintained in Dulbecco’s modified Eagle’s Medium (DMEM) with 10% FBS at 37 °C and 5% CO_2_) were transiently transfected using Lipofectamine 2000 (Life Technologies, NY, USA). Retroviral particles were obtained and concentrated by centrifugation of the conditioned media (6000× *g* overnight at +4 °C). LPS-stimulated B cells were transduced by retroviral pool spin infection in 8 µg/µL of polybrene (Sigma-Aldrich, St. Louise, MO, USA). Each retroviral pool contained vectors expressing WT MYC, T58A MYC or T58I MYC as well as both BCL2L1 and BMI1. Cells were re-plated in complete culture medium supplemented with 2 µg/mL doxycycline.

### 4.4. Cell Culture and Dose Dependent Expression of MYC

All three transduced cell lines (WT MYC, T58A MYC and T58I MYC) were cultured in GlutaMAX-RPMI (Gibco, NY, USA) supplemented with 10% FBS (0.01 M HEPES, 1× sodium pyruvate, 0.05 mM 2-beta-mercaptoethanol (Gibco) and 0.1 mg/mL penicillin-streptomycin (Sigma) with the addition of 2 µg/mL doxycycline. For titration experiments, each cell line was removed from doxycycline medium, washed with PBS (3×) and RPMI (3×), and subsequently cultured under 7 different concentrations of doxycycline (0, 25, 50, 100, 300, 600 and 1000 ng/mL) for 48, 96 or 144 h at 37 °C and 5% CO_2_.

### 4.5. Flow Cytometry

Cells expressing WT MYC, T58A MYC or T58I MYC were stained for CD19, CD45R and CD90 markers using monoclonal antibodies, as described previously [3]. Cell cycle analysis was performed with propidium iodide staining, as described previously [46]. Relative cell size was quantified based on the mean fluorescence intensity of the forward scatter. Cell marker and cell cycle samples were analyzed using a BD LSRFortessa Cell Analyzer (BD Biosciences, San Jose, CA, USA), and acquired data were passed to FlowJo software (v10, Tree Star, Inc. OR, USA) for data analysis.

### 4.6. Western Blot

A total of 10^6^ cells were collected and directly lysed using 4× LDS sample buffer (Thermo Fisher Scientific, TMO, NY, USA) prior to loading onto SDS-PAGE (4–12% Tris-Glycine) gels. Western blot reagents, including gels and nitrocellulose membranes for the iBlot2 Dry-blotting system, were purchased from Life Technologies. Anti-actin (1:100,000, Sigma) and anti c-MYC (1:1000, 9 × 10^10^, Thermo Fisher Scientific) antibodies were used to detect proteins.

### 4.7. RNA Isolation, Library Preparation and RNA-Sequencing

Total RNA was isolated using the RNeasy kit (Qiagen, California, USA), and libraries for NGS were prepared using the TruSeq2.0 sample preparation kit (Illumina, San Diego, CA, USA) according to the manufacturer’s instructions and included a poly-A selection step using poly-T oligo attached magnetic beads. The libraries were thereafter multiplexed on an Illumina HiSeq2000 instrument generating on average 28.6 million 50 bp single-end reads per sample.

### 4.8. RNA-Seq Alignment

Read quality was assessed by FastQC (v.0.10.1). The mouse reference genome, build GRCm38.87, was obtained from the ensemble repository with the corresponding gtf annotation file. The reference genome was concatenated with genomic sequences for the protein coding sequences of the four genes present in the retroviral vectors used in the study; human *MYC*, *BCL2L1* and *BMI1* as well as *EGFP* and subsequently indexed using STAR (v.2.5.1b). The 50 bp reads were then aligned to the mouse genome using the splice-aware short read aligner STAR with default options and -outFilterMultimapNmax set to 1 and --sjdbGTFfile pointing to the GRCm38.87 gtf file appended with lines corresponding to the four added features [47].

### 4.9. Gene Expression and Gene Cluster Profile Analyses

Reads per feature (concatenated exons for one gene) were obtained using the featurecounts function from the subread package (v1.5.2) with default options and -g set to “gene_id” and -t set to “exon” [48]. The count table was imported into the R environment (v.3.5.0) and genes with counts per million ≥ 0.95 for at least 3 samples were included in the analysis (*n* = 14,142). Tests for differential expression were conducted using the Bioconductor (v3.8) package edgeR (v.3.24.3) using the glmQLF framework [49]. The differentially expressed features were subsequently subject to hierarchical clustering (Euclidian distances and Ward’s method) using mean log_2_ transformed RPKM values for each dose and genotype. Functional classification of the clustered features was conducted by means of enrichment tests within the databases for GO Biological Processes and KEGG pathways using clusterProfiler (v.3.10.1) [50].

### 4.10. Distances to Canonical E-Boxes

The mouse reference genome GRCm38.87 was indexed for Bowtie1 (v1.2.0) [51] followed by alignment of the canonical E-box 6-mer motif (CACGTG) with default options plus the -a flag and -v set to 0, yielding 536,520 uniquely aligning motifs. After conversion of the output file via .bam to .bed using samtools (v1.6) [52] and BEDtools (v2.27.1, https://bedtools.readthedocs.io), respectively, the distances to the closest known feature for each E-box location were derived using the annotatepeaks.pl script from the Homer package for features in the GRCm38.87 .gtf file (http://homer.ucsd.edu/homer/ngs/annotation.html).

### 4.11. Overlap with Published Datasets

Appendix A from Sabò et al. (*Nature*, 2014) [18] was downloaded and filtered for genes with significant transcript level changes between any of the conditions (FDR q-value ≤ 0.05). The resulting list of genes was subsequently used to look for significant overlaps with all the significantly changed genes for WT MYC from the present study as well as overlaps with each of the twelve clusters. Fisher’s exact tests were used to assess the significance of overlaps.

A list containing the 100 most significantly regulated genes upon attenuation of MYC expression from Muhar et al. (*Science*, 2018) [21] was downloaded from Appendix A in that article. The list of human genes was subject to orthology conversion using the biomaRt Bioconductor package. Eighty-eight of the 100 genes could be converted to a unique mouse identifier, and these were subsequently used for comparisons with the present study. A similar approach was taken for a set of 668 direct MYC target genes published by Zeller et al. (PNAS 2006) [20] and presented in Appendix A.

A list of LPS-regulated genes (*n* = 4140) in normal mouse B-cells that represent part of the normal gene regulation program of MYC were taken from the embr201947987-sup-0004-datasetev2.xlsx supplementary file from Tesi et al. [17]. Fisher’s exact tests (one-tailed) were used to assess the extent of significant over-representation of the number of genes in overlaps between these genes and sets of genes from this study.

### 4.12. Gene set Enrichment Analysis

A ranked list of all genes included in the analysis based on their sensitivity to MYC level in relation to mutational status was generated by taking the ratio of change in fold change (FC_dox 25/0_/FC_dox 600/300_) between the mutants (T58A/T58I). The pre-ranked list was used for gene set enrichment analysis using the R package fGSEA(v1.8.0) with gene sets downloaded from the MSigDb (http://software.broadinstitute.org/gsea/msigdb/index.jsp) for the KEGG [53] pathway database (c2.cp.kegg.v6.2.symbols.gmt), gene ontology biological processes (c5.bp.v6.2.symbols.gmt) and hallmark gene sets (h.all.v6.2.symbols.gmt) [25].

### 4.13. Protein Conformation Prediction

Protein conformation characteristics were predicted using web-based prediction software as follows. Alpha-helical propensity was predicted using the Agadir predictor (http://agadir.crg.es) [54]. Protein backbone dynamics were predicted using the Dynamine 3.0 predictor (http://dynamine.ibsquare.be) [55]. Protein interaction propensity was predicted using the ANCHOR predictor (https://iupred2a.elte.hu) [56]. All three prediction sets were performed using the designated webservers on 4 July 2018.

## 5. Conclusions

We have used a primary B-cell model to determine how progressively increased MYC levels drive cells, which are otherwise predisposed for oncogenic transformation, into a lymphoma phenotype, as well as how this process is augmented by lymphoma-associated mutations affecting the MYC protein. Regulated genes include normal MYC target genes, but abnormal repertoires of regulated genes were also identified, particularly among genes that are differentially regulated by mutant MYC compared to wild type MYC. These genes show a significant tendency to be involved in cell cycle-related processes such as DNA replication and G_2_ to mitosis transition. The T58A and T58I lymphoma-associated mutations affect MYC function differently. While T58A increases the sensitivity of MYC responses leading to cell growth and proliferation, gene regulation and the measured phenotypic responses tend to be less sensitive in T58I compared to wild type MYC. The explanation for the T58I effects could be that it is important to optimize MYC activity in order to maximize the induction of cell growth and proliferation without excessive induction of apoptosis or cellular senescence. Thus, in cells that have functional MYC-induced apoptosis and/or cellular senescence processes, one might envisage an oncogenic role for MYC mutations, such as T58I, that reduce the level of oncogenic MYC activation. On the other hand, in cells where apoptotic and/or senescence processes have been ablated by other oncogenic events, mutations such as T58A that enhance MYC activity would be favored. Having access to this kind of reasoning about the possible order of events leading to different cases of lymphoma might give important clinical insights. Alternatively, T58I could enhance lymphogenic processes distinct from cell growth and proliferation. Unlike T58A, T58I is predicted to increase the rigidity of the local peptide backbone and to decrease protein action propensity. It is possible that such protein conformation effects could result in reduced or alternatively altered protein binding activity within the MYC-box 1 region of the MYC N-terminus. As yet hypothetical therapeutic agents that altered the protein conformation of the region of MYC surrounding threonine 58 could thus potentially be used to increase or decrease MYC activity.

## Figures and Tables

**Figure 1 cancers-13-06093-f001:**
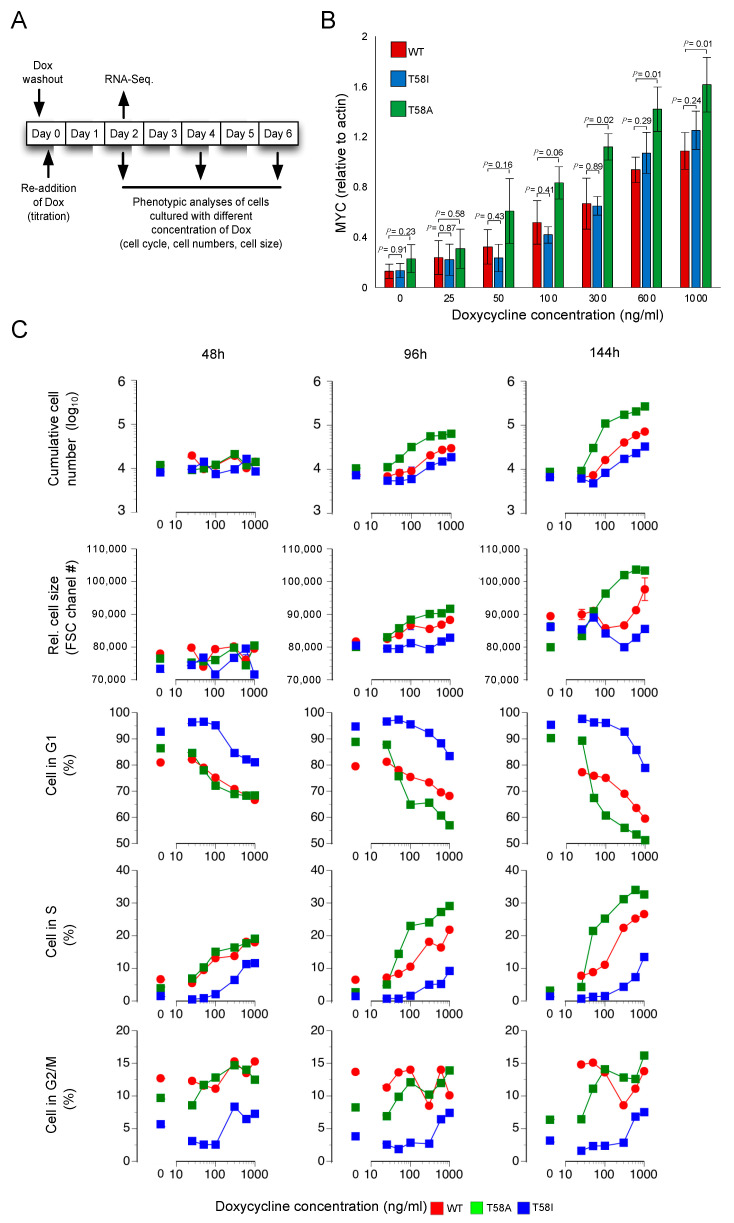
Phenotypic characterization of immortal B-cells expressing doxycycline regulatable MYC. (**A**) Experimental setup, time points for doxycycline wash out and re-addition as well as time points for data collection and RNA sampling (**B**) MYC protein levels at different levels of doxycycline in culture medium (0, 25, 50, 100, 300, 600 and 1000 ng/mL) assessed by Western Blot for WT MYC (red), T58A MYC (green) and T58I MYC (blue) in relation to actin levels. Mean ± SD (*n* = 3) is shown. (**C**) Flow cytometry data for WT MYC (red), T58A (green) and T58I (blue) sampled at 48, 96 and 144 h following addition of doxycycline at 0, 25, 50, 100, 300, 600 and 1000 ng/mL. From top to bottom, the panels represent: cumulative live cell number, relative cell size estimated by the forward scatter channel (FSC) and G_0_/G_1_, S and G_2_/M cell cycle stage assessment for cells stained with propidium iodide (percentages). The data are from a single experiment. Similar data for the 48 h time point (used for RNA sequencing experiments) were obtained in a separate experiment.

**Figure 2 cancers-13-06093-f002:**
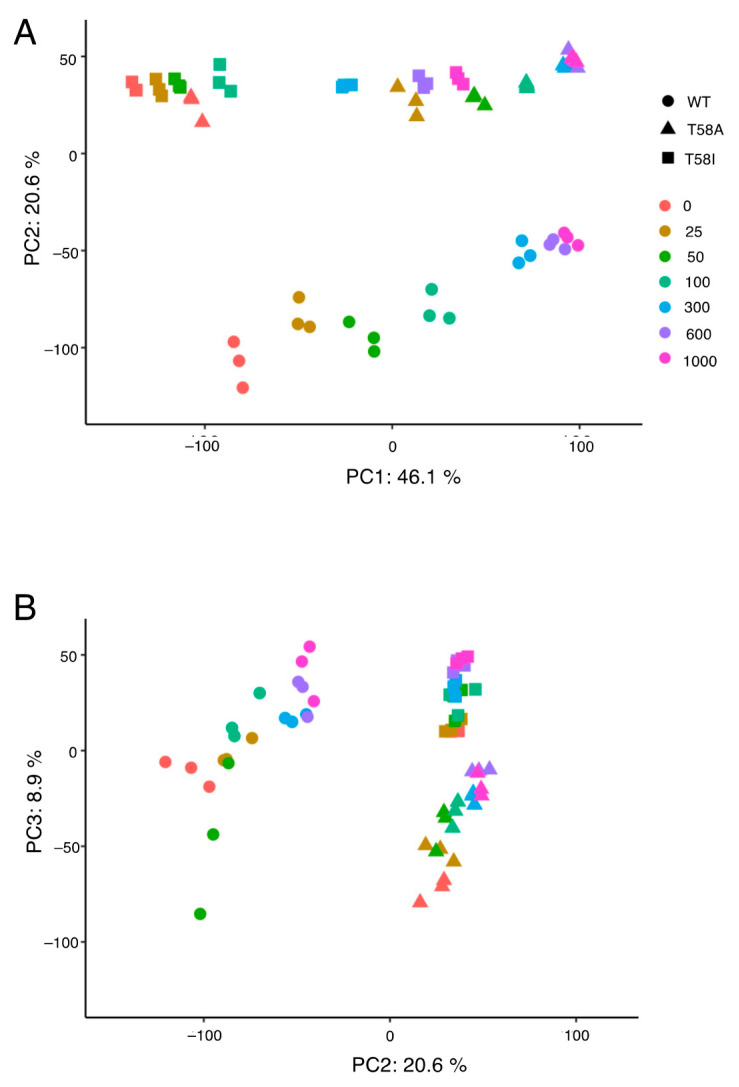
Principal component analysis of global gene expression data. The first three principal components describe 75.6% of the variation in the global gene expression data across the three MYC genotypes and seven expression levels. Levels of MYC WT (circles), MYC T58A (triangles) and MYC T58I (squares) were controlled by titrating in doxycycline at the indicated concentrations (ng/mL, colors). (**A**) Scatter plot of principal component (PC) 1 versus PC2. (**B**) Scatter plot of PC2 versus PC3. The proportion (%) of the variation accounted for by each PC is shown.

**Figure 3 cancers-13-06093-f003:**
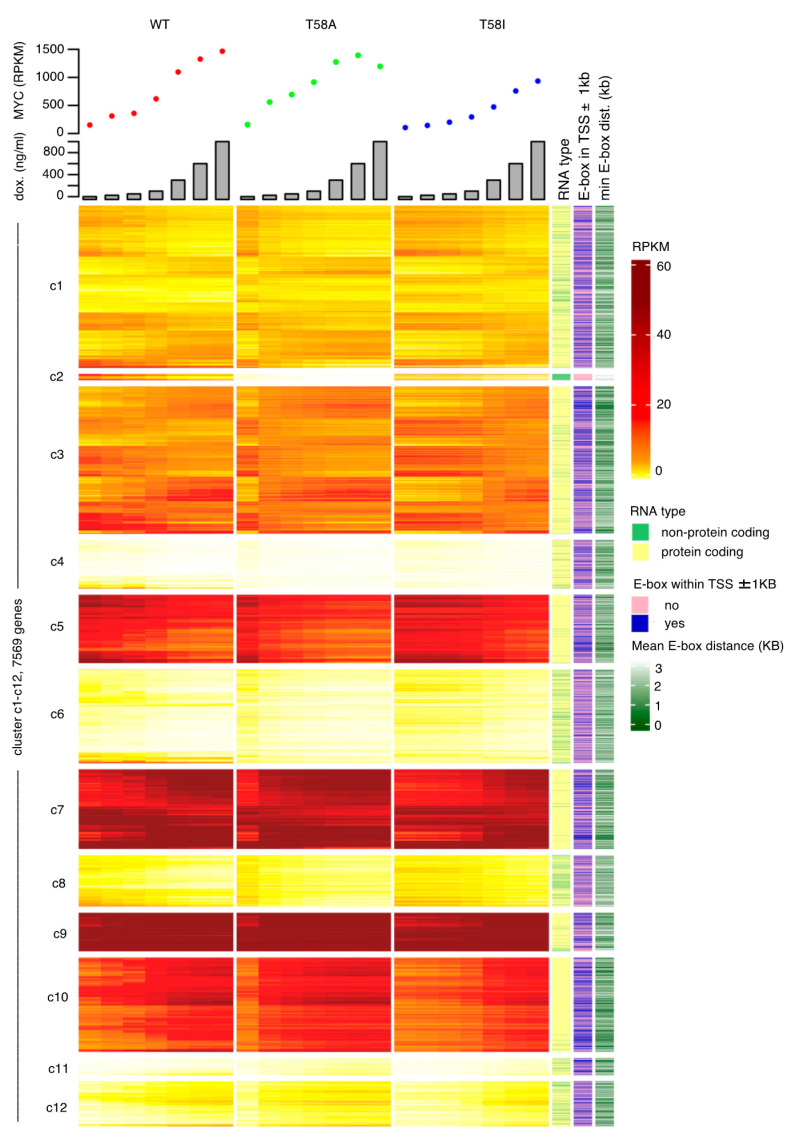
Expression levels for gene clusters. Hierarchical clustering based on expression levels for the union of genes that were either significantly changed upon increased MYC levels in WT or that were differentially regulated in response to increasing MYC levels between MYC WT and MYC T58A or between MYC WT and MYC T58I (7569 genes) yielded 12 clusters (*n* = 3 independent experiments). The heatmap shows reads per kilobase per million of reads (RPKM) for all clustered genes (rows) for different levels (plotted above the heat map) of the three MYC genotypes (MYC WT (red dots), MYC T58A (green) and MYC T58I (blue), columns). Doxycycline concentrations (ng/mL) used to treat the cells are indicated by the bar-chart above the main heatmap. To the right of the main heatmap are three tracks describing, from left to right, whether the differentially regulated genes encode for protein-coding (yellow) or non-coding (light green) RNA, whether (blue) or not (pink) a gene has a canonical E-box motif within 1kb of its transcription start site (TSS), and the last track describes the distance between the TSS and the closest canonical E-box motif.

**Figure 4 cancers-13-06093-f004:**
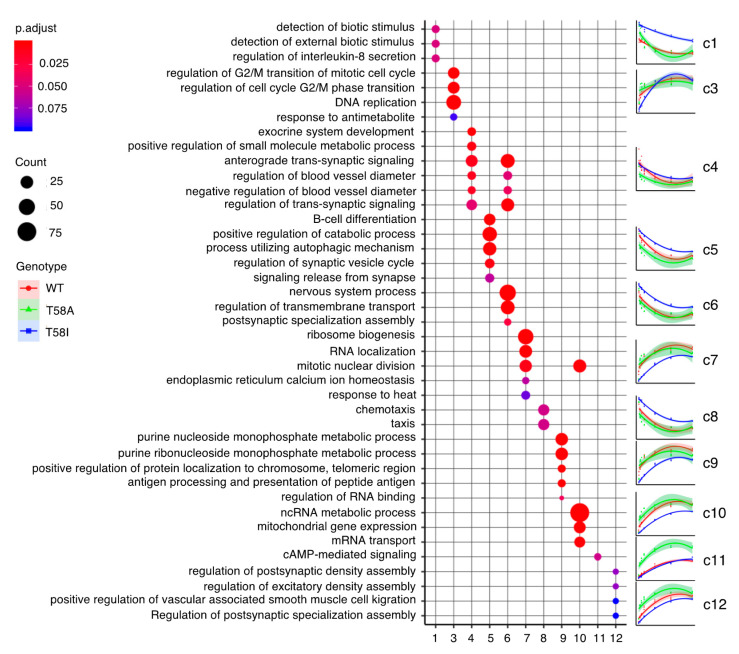
Gene ontology functional classification of clustered genes. Over-representation tests of clustered genes in gene sets with different gene ontology (GO) terms, representing biological processes, for clusters c1 and c3–c12. The most enriched gene sets following GO term simplification are shown. Median RPKM values for genes in denoted clusters are plotted (right panels) for MYC WT (red), MYC T58A (green) and MYC T58I (blue) for increasing doxycycline levels (0, 25, 50, 100, 300, 600 and 1000 ng/mL), and the colored lines corresponding to the respective MYC genotype represent a 2nd degree polynomial regression model where the shaded areas indicate the 95% confidence interval for the model (larger high resolution plots are available in Appendix A).

**Figure 5 cancers-13-06093-f005:**
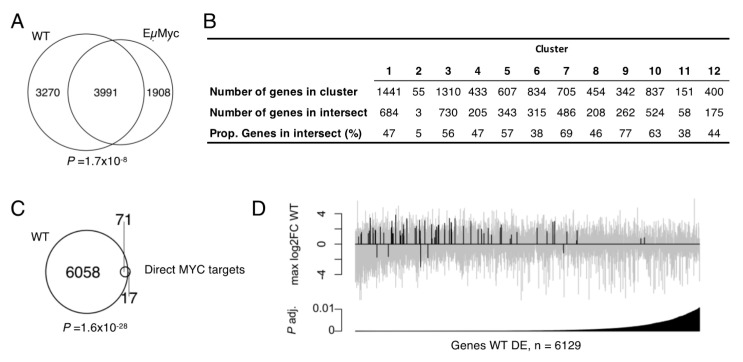
Comparisons with published datasets relevant for MYC-driven lymphogenesis. (**A**) Overlap between differentially regulated genes in WT MYC and significantly changed genes between any of the conditions measured by Sabò et al. in the Eµ-Myc mouse model [18]. *p*-value from Fisher’s exact test. (**B**) Representation of significantly changed genes in the Sabò et al. dataset in gene clusters c1–c12. Genes from clusters c3, c7, c9 and c10 were over-represented in the regulated genes from the Sabò et al. study (Fisher’s exact test, *p* = 0.02, 6.0 × 10^−20^, 2.0 × 10^−20^ and 2.0 × 10^−9^, respectively). (**C**) Overlap between 88 of the 100 most significant direct MYC target genes derived from Muhar et al. [21] that could be converted to the mouse ortholog with a unique gene identifier and genes with significantly altered transcript levels in WT MYC. *p*-value from Fisher’s exact test. (**D**) A total of 6129 genes that were differentially regulated in WT MYC and that had unique gene identifiers following ortholog conversion between human and mouse ranked by adjusted *p*-value. Each grey bar represents one gene with the 71 genes that were also present in the data from Muhar et al., marked in black. The set of 71 genes was significantly enriched in the set of most significantly regulated genes in WT MYC (Fisher’s exact test, *p*-value = 0.0012).

**Figure 6 cancers-13-06093-f006:**
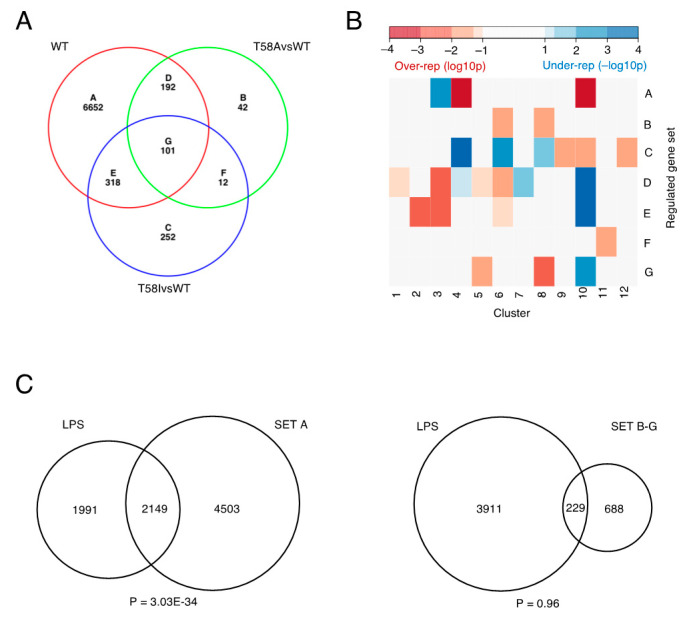
Differences in differentially regulated genes between MYC WT, MYC T58A and MYC T58I. (**A**) Venn diagram representation of genes with significant transcript level changes (FDR q-value ≤ 0.01 and max fold change ≥ 2). The diagram shows the set of genes with significant regulation in MYC WT (WT) and its overlap with the sets of genes that are regulated significantly differently from WT MYC in MYC T58A (T58A vs. WT) and/or MYC T58I (T58I vs. WT). (**B**) Heatmap representation of Fisher’s exact test significance levels for intersects between the Venn diagram subsets in panel A (A–G) and genes in clusters 1–12. Intersects with a significantly larger overlap than expected are represented in red (over-rep) and intersects with an overlap significantly smaller than expected are represented in blue (under-rep). (**C**) Comparison differentially regulated gene sets (A and B–G) with lipopolysaccharide (LPS) regulated genes, representing normal MYC target genes. The names of LPS-regulated genes were obtained from previously published work [17]; supplementary data file “embr201947987-sup-0004-datasetev2.xlsx”, all groups. Venn diagrams show a significant overlap between Set A genes and LPS-regulated genes and a smaller overlap for Set B–G genes (all sets except for Set A) that is not statistically significant. *p*-values to test over-representation of genes in the overlaps are shown (Fisher’s exact test, one-tailed).

**Figure 7 cancers-13-06093-f007:**
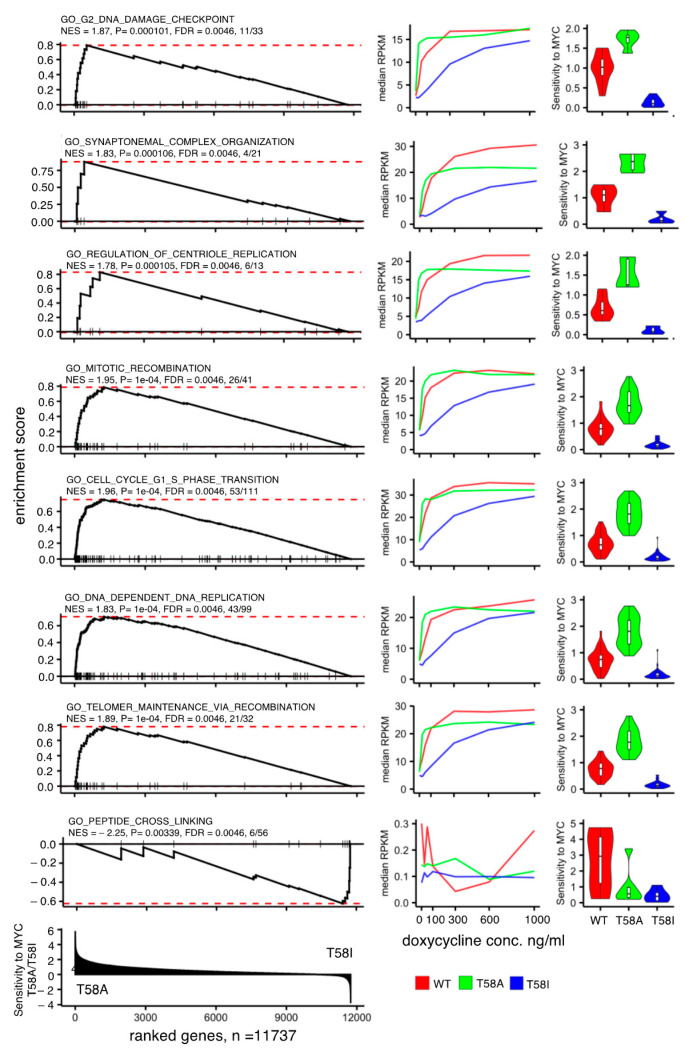
Enriched pathways for genes with sensitivity differences to the levels of MYC T58A and MYC T58I. Representative examples of gene sets that are significantly enriched in a gene set enrichment analysis (GSEA) of genes ranked according to the extent of their sensitivity to T58A MYC in relation to T58I MYC. Left panels: GSEA plots showing a running sum statistic that increases when genes in the ranked list are present in the functional gene set tested. The position of these genes in the rank is indicated by vertical lines. The red dashed line shows the enrichment score value (ES, maximum value of the running sum score). The analysis was performed on 11,737 genes for which human orthologues could reliably be identified and the sensitivity-based rank is shown in the lowest plot. The pathway name is indicated above each plot followed by the normalized enrichment score (NES), GSEA *p*-value, false discovery rate adjusted *p*-value (FDR) and the number of genes in the GSEA leading edge (interval between the start of the rank and the gene where ES is attained) in relation to the number of genes in the pathway. Middle panel: Expression-level plots showing median reads per kilobase per millions of reads (RPKM) values for leading edge set of genes identified for each GSEA gene set. Median RPKM values for WT MYC (red), T58A MYC (green) and T58I MYC (blue) at different doxycycline levels (0, 25, 50, 100, 300, 600 and 1000 ng/mL) are plotted. Right panels: Violin plots showing the MYC-level sensitivity of the leading-edge genes in the indicated pathway for WT MYC (red), T58A MYC (green) and T58I MYC (blue). Sensitivity was determined by the change in absolute fold change between the fold change between 0 and 25 ng/mL and the fold change between 300 and 600 ng/mL of doxycycline (FC_25/0_/FC_600/300_).

**Figure 8 cancers-13-06093-f008:**
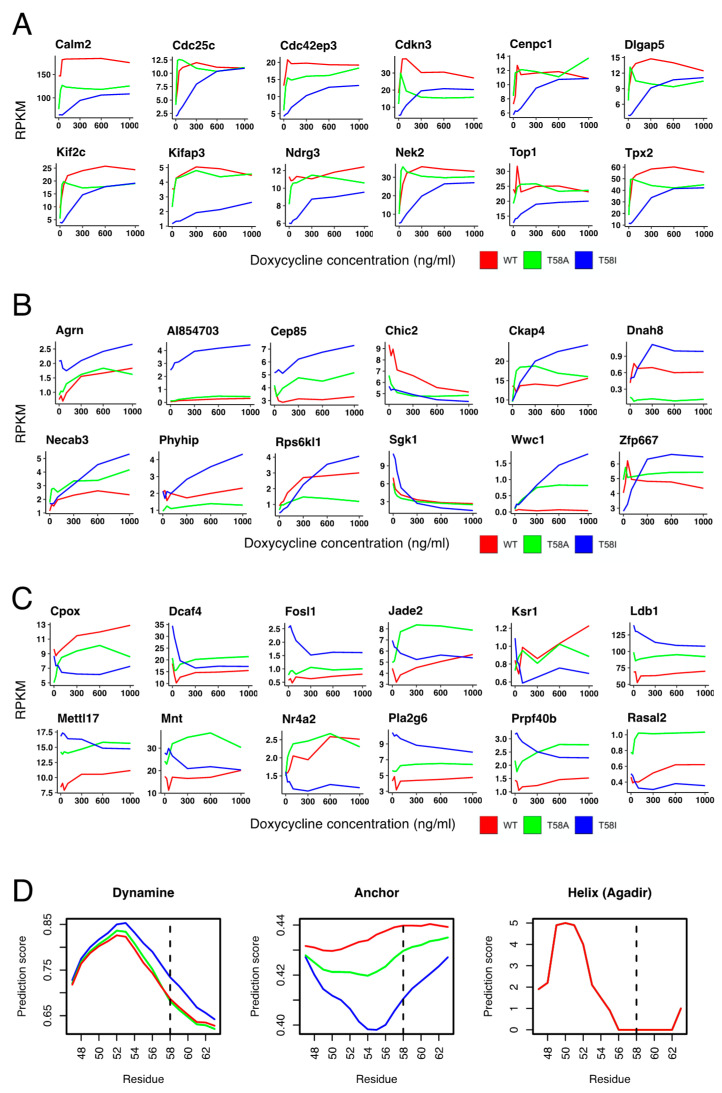
Genes that are differentially regulated by T58I MYC. (**A**) Plots showing the expression levels (RPKM) as a function of doxycycline concentration (0, 25, 50, 100, 300, 600 and 1000 ng/mL) for WT MYC (red), T58A MYC (green) and T58I MYC (blue). (**A**) Representative genes that show lower regulation by T58I MYC than WT MYC and/or T58A MYC. (**B**) Representative genes that show higher regulation by T58I MYC than WT MYC and/or T58A MYC. (**C**) Representative genes that show a different regulation direction by T58I MYC than WT MYC and/or T58A MYC. (**D**) Protein conformation predictions for the conserved MYC-Box I region, containing residue 58 (dashed line), in WT MYC (red), T58A MYC (green) and T58I MYC (blue). Dynamine predicts rigidity of the protein backbone (scale 0–1, where 1 is highly rigid). Anchor predicts protein interaction propensity (scale 0–1, where 1 is the highest propensity level). Helix (Agadir) is a prediction of alpha-helicity using the Agadir algorithm (the results are essentially identical for the three proteins, explaining the existence of only one visible plotted line).

**Table 1 cancers-13-06093-t001:** Quantitative characteristics of MYC responsive gene clusters, clustered according to RPKM, MYC response and MYC mutant status.

Cluster	1	2	3	4	5	6	7	8	9	10	11	12
Gene number (*n*)	1441	55	1310	433	607	834	705	454	342	837	151	400
Overall reg. pattern ^a^	down	down	up	down	down	down	up	down	up	up	up	up
Up-reg genes (%) ^b^	39	0	51	10	2	7	68	2	73	98	92	97
Down-reg genes (%) ^b^	61	100	49	90	98	93	32	98	27	2	8	3
Expression level (RPKM) ^c^	3.1	1.2	8.3	0.1	19.9	0.4	46.2	1.1	128.1	18.7	0.2	1.0
Non-coding genes (%) ^d^	15	95	6	10	3	25	2	27	7	2	25	25
Median gene length (bp) ^e^	4133	366	4204	4906	4390	3412	3099	3349	2438	3652	4253	3498
Genes-E-box close to TSS (%) ^d^	46	5	50	43	48	37	53	39	48	56	52	47
Median min. dist. TSS–E-box) ^e^	885	5081	717	1115	837	1333	659	1142	666	574	779	913

^a^ Regulation direction of genes representing > 50% of genes; ^b^ proportion of genes with alternative regulation directions: up = RPKM(1000 ng/mL dox) > RPKM(0 ng/mL dox): down = RPKM(1000 ng/mL dox) < RPKM(0 ng/mL dox); ^c^ median RPKM value; ^d^ over-represented (red) or under-represented (blue), chi-squared test, adjusted *p*-value < 0.05; ^e^ length (bp) significantly longer (blue) or shorter (red) than for mean length for non-regulated genes, 1000-fold random resampling of *n* non-regulated genes, adjusted *p*-value < 0.02. Abbreviations: reg—regulation/regulated, RPKM—reads per kilobase per million reads, bp—base pairs, TSS—transcription start site, min. dist.—minimum distance in base pairs.

## Data Availability

The sequencing data generated and analyzed in the present study can be accessed via the gene expression omnibus through accession number: GSE122602.

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
