# Peer review of "Differential Transcriptional Reprogramming by Wild Type and Lymphoma-Associated Mutant MYC Proteins as B-Cells Convert to a Lymphoma Phenotype"

_cancers, 2021, doi:10.3390/cancers13236093_

Round 1

Reviewer 1 Report

The authors used here murine B cells trasnduced with retroviral vectors containing coding sequence for WT MYC or one of 2 different MYC mutants (T58I and T58A). Using these cells, the authors characterized the transcriptional effects of increasing MYC levels and determined how this gene regulation program is modified.

This manuscrit is well written, very interesting and complete.

Minor changes are required:

figure 2 is too small and should be enlarged.

As said previously, this study is very intersting and helps to understand how the mutations in MYC gene drive oncogenesis. However, it would be interesting to link all these interesting results to clinical/biological practice. 

Reviewer 2 Report

Mahani and coll. report very interesting results about the MYC-induced transcriptional reprograming of B-cells and their subsequent transition to a B-cell lymphoma phenotype.

Overall, I feel that the authors have delivered a technically solid piece of work and applaud their efforts to make this work understandable for the greatest number.

I have only minor points about this manuscript:

Supplementary Figure S1: I do not understand how/why CD45R is elevated in experimental T58A/T58I conditions vs WT?

Statistical differences  should be  indicated such as, for example, in Figure 1B.

Figure 1B: Periods rather than comas on the Y axis.

The number of experiments should be indicated for Figure 1.

Even if described in a previous study (Eur J Immunol 2019), more experimental details would be of interest in section 4.1 concerning, for example, mouse genotype, percentage of B-cells in the freshly isolated splenocyte population, transfection efficiency …
